# Maternal Supplementation with N-Acetylcysteine Modulates the Microbiota-Gut-Brain Axis in Offspring of the Poly I:C Rat Model of Schizophrenia

**DOI:** 10.3390/antiox12040970

**Published:** 2023-04-20

**Authors:** Diego Romero-Miguel, Marta Casquero-Veiga, Javier Fernández, Nicolás Lamanna-Rama, Vanessa Gómez-Rangel, Carlos Gálvez-Robleño, Cristina Santa-Marta, Claudio J. Villar, Felipe Lombó, Raquel Abalo, Manuel Desco, María Luisa Soto-Montenegro

**Affiliations:** 1Instituto de Investigación Sanitaria Gregorio Marañón, 28007 Madrid, Spain; 2Grupo de Investigación “Biotechnology in Nutraceuticals and Bioactive Compounds-BIONUC”, Departamento de Biología Funcional, Área de Microbiología, Universidad de Oviedo, 33006 Oviedo, Spain; 3Instituto Universitario de Oncología del Principado de Asturias (IUOPA), 33006 Oviedo, Spain; 4Instituto de Investigación Sanitaria del Principado de Asturias (ISPA), 33011 Oviedo, Spain; 5Departamento de Bioingeniería, Universidad Carlos III de Madrid, 28911 Madrid, Spain; 6Grupo de Investigación de Alto Rendimiento en Fisiopatología y Farmacología del Sistema Digestivo (NeuGut-URJC), Universidad Rey Juan Carlos, 28922 Alcorcón, Spain; 7Departamento de Ciencias Básicas de la Salud, Universidad Rey Juan Carlos (URJC), 28922 Alcorcón, Spain; 8Departamento de Física Matemática y de Fluidos, Universidad Nacional de Educación a Distancia (UNED), 28040 Madrid, Spain; 9Grupo de Trabajo de Ciencias Básicas en Dolor y Analgesia, Sociedad Española del Dolor (SED), 28046 Madrid, Spain; 10Grupo de Trabajo de Cannabinoides, Sociedad Española del Dolor (SED), 28046 Madrid, Spain; 11Unidad Asociada I+D+i del Instituto de Química Medica (IQM), Consejo Superior de Investigaciones Científicas (CSIC), 28006 Madrid, Spain; 12Centro de Investigación Biomédica en Red de Salud Mental (CIBERSAM), 28029 Madrid, Spain; 13Centro Nacional de Investigaciones Cardiovasculares (CNIC), 28029 Madrid, Spain

**Keywords:** Poly I:C, schizophrenia, microbiota-gut-brain axis, inflammation, oxidative stress

## Abstract

The microbiota-gut-brain axis is a complex interconnected system altered in schizophrenia. The antioxidant N-acetylcysteine (NAC) has been proposed as an adjunctive therapy to antipsychotics in clinical trials, but its role in the microbiota-gut-brain axis has not been sufficiently explored. We aimed to describe the effect of NAC administration during pregnancy on the gut-brain axis in the offspring from the maternal immune stimulation (MIS) animal model of schizophrenia. Pregnant Wistar rats were treated with PolyI:C/Saline. Six groups of animals were studied according to the study factors: phenotype (Saline, MIS) and treatment (no NAC, NAC 7 days, NAC 21 days). Offspring were subjected to the novel object recognition test and were scanned using MRI. Caecum contents were used for metagenomics 16S rRNA sequencing. NAC treatment prevented hippocampal volume reduction and long-term memory deficits in MIS-offspring. In addition, MIS-animals showed lower bacterial richness, which was prevented by NAC. Moreover, NAC7/NAC21 treatments resulted in a reduction of proinflammatory taxons in MIS-animals and an increase in taxa known to produce anti-inflammatory metabolites. Early approaches, like this one, with anti-inflammatory/anti-oxidative compounds, especially in neurodevelopmental disorders with an inflammatory/oxidative basis, may be useful in modulating bacterial microbiota, hippocampal size, as well as hippocampal-based memory impairments.

## 1. Introduction

The microbiota-gut-brain axis is a complex interconnected system that appears to be altered in the schizophrenia (SCZ) [1]. Certain psychiatric disorders have purportedly not only brain-related but also systemic causes, based on an inflammatory response of the organism in which bacterial components of the gut microbiota may participate [2]. The alteration of the bidirectional homeostasis between the intestine and the brain, which involves neurological, metabolic, hormonal, and immunological signaling pathways, can favor the appearance of metabolic, immunological, or CNS disorders, with behavioral changes, among other clinical manifestations. In this sense, first episode psychosis (FEP) patients show gut microbiota alterations with differences in the composition and number of microbes, which in turn are associated with their symptoms severity [3]. Furthermore, patients with acute SCZ have shown no changes in microbiota richness but decreased microbial biodiversity [4], indicating an altered gut microbiota.

Serological markers of gut bacterial translocation have also been reported to be substantially increased in subjects with psychosis and significantly correlated with systemic inflammatory markers [5]. In turn, cytokine levels correlate with the severity of clinical symptoms, and it has been suggested that the resulting neuroinflammation is directly involved in the pathogenesis of SCZ [6]. Similarly, gut microbiota changes have been observed in different animal models of psychiatric disorders, including autism and SCZ [7,8]. In this regard, the maternal immune stimulation (MIS) model is one of the most widely used animal models for SCZ. This is a well-validated animal model produced by maternal exposure to a viral infection with Polyinosinic:polycytidylic acid (Poly I:C), a viral-mimicking agent which stimulates toll-like receptor 3 (TLR3) during pregnancy. The MIS model is based on the known association between maternal viral infection during pregnancy and increased risk of the onset of SCZ in offspring [9,10,11]. Prenatal exposure to Poly I:C leads to the development of a broad schizophrenic spectrum that includes behavioral, neurological, immunological, structural and metabolic deficits, mainly characterized by behavioral and in vivo imaging techniques [10,11,12,13,14]. However, the impact of Poly I:C on offspring’s gut microbiota has been barely explored. Only a few studies have revealed the microbiota composition in the MIS model [7,15,16]. MIS model studies carried out in rats show an increase in specific bacterial taxa, such as *E. coli*, *Lactobacillus* spp., *Bifidobacterium* spp. and *Bacteroides* spp. [16], which is consistent with clinical studies [3,17]. In mice, the MIS model has failed to show differences in alpha diversity (calculated as Shannon index) [15] but detected large differences in the composition of gut bacteria between MIS offspring and control animals, mainly in the bacterial classes *Clostridia* and *Bacteroidia*, with an increase in bacterial families, such as *Lachnospiraceae* and *Prevotellaceae* [7]. Although these changes could suggest differences in the *Firmicutes/Bacteroidetes* ratio, significant differences were not found between MIS and control animals with respect to this ratio [15].

It is important to note that gut microbiota undergoes important maturation and development processes throughout life, being the perinatal, postnatal and adolescence stages critical periods for its progress [18]. In this sense, manipulation of the gut microbiota-brain axis through diet in these critical time windows appears to be an innovative therapeutic tool to prevent the risk of neurodevelopmental disorders that may affect lifelong health. In this regard, the likely involvement of inflammation/oxidative stress (IOS) in the pathophysiology of SCZ has raised the potential use of anti-oxidant and anti-inflammatory (anti-IOS) compounds as therapeutic strategies in this disorder [19]. Among them, the use of N-acetylcysteine (NAC) has received particular attention. NAC, a precursor of the amino acid L-cysteine (L-Cys), has remarkable antioxidant properties as it increases the availability of L-Cys, facilitating the synthesis of glutathione (GSH), which is the main non-enzymatic antioxidant defense that protects cells from reactive oxygen species [20,21]. This ability to regulate GSH biosynthesis is the key to its therapeutic efficacy. NAC is also able to reduce proinflammatory cytokines, such as IL-6, TNF-α and IL-1β [21]. In clinical trials, NAC is often administered as adjunctive treatment to conventional treatment with antipsychotics [22]. Overall, there is evidence that NAC has beneficial effects on SCZ symptoms beyond those induced by antipsychotics alone [22,23]. However, to our knowledge, no study has evaluated the effects of NAC on the interconnection between gut microbiota and the brain in SCZ. In this regard, few studies have demonstrated that NAC therapy may alleviate gut dysbiosis in several pathologies, such as diabetes and metabolic diseases [24,25,26], hypertension [27], and cardiovascular disease [28]. Moreover, three recent studies have evaluated maternal NAC treatment via regulating the gut microecology to alleviate maternal placental oxidative stress and inflammatory response in sows [29,30] and to prevent hypertension in spontaneously hypertensive rat offspring [27], which highlights its possible use in other pathologies with inflammatory and neurodevelopmental basis, such as SCZ.

Thus, this study aimed to investigate the composition, taxonomy, and functional diversity of the gut microbiota in an MIS animal model of SCZ and to evaluate the anti-IOS potential of NAC in the reversion of the gut microbiota dysbiosis produced in the MIS model. In addition, this study benefits from the study of the changes in hippocampal volume and hippocampal-based memory, measured by magnetic resonance imaging and novel object recognition (NOR) task, respectively, as well as the association between gut microbiota and hippocampal volume.

## 2. Materials and Methods

### 2.1. Animals

Seventy-two male Wistar rats were maintained at constant temperature (24 ± 0.5 °C) under a 12-h light/dark cycle, with free access to chow/water. All animal procedures were conducted in conformity with the European Communities Council Directive 2010/63/EU, following the ARRIVE guidelines [31] and approved by the Ethics Committee for Animal Experimentation of Universidad Rey Juan Carlos and Gregorio Marañón Hospital (ES280790000087). Two batches of animals were used: 44 animals underwent behavioral studies, and 69 animals underwent imaging and metagenomics. The number of animals per cohort was calculated for the 6 experimental groups, considering the 3Rs principle in animal experimentation and assuming independent groups, the detection of large effect (1 SD), and a power of 80%.

### 2.2. Treatments

MIS model: On gestational day 15 (GD15), Poly I:C (4 mg/kg, Sigma-Aldrich, Madrid, Spain) or saline solution was administered i.v. to pregnant Wistar rats. This protocol is associated with a high risk of the onset of behavioral and neurological deficits in the offspring at adulthood, similar to those found in patients with SCZ [11,14]. On post-natal day (PND) 21, male-offspring were weaned and housed 2–4 per cage. Females were not studied because they do not reproduce the well-characterized SCZ-like deficits in males of the MIS model at PND100 [32].

NAC treatment during gestation: NAC (500 mg/kg, Sigma-Aldrich, Madrid, Spain) diluted in drinking water (VH) was administered at two different time periods during gestation: throughout gestation (NAC21, 21 days of administration) or after the Poly I:C/saline administration (NAC7, 7 days of administration, from gestational days 15 to 21). Water intake was monitored to calculate the dose of NAC ingested by animals and freshly prepared to prevent NAC degradation. The dose of NAC ingested daily by each animal was similar in all groups (minimum dose: 518 mg/kg, maximum dose: 552 mg/kg).

Animals were divided into six groups (6–12 animals per group) according to the factors of the study: phenotype (Saline, MIS) and treatment (water without NAC, water with NAC for 7 days–NAC7, water with NAC for 21 days–NAC21).

### 2.3. Novel Object Recognition (NOR) Test

A behavioral test was performed at PND70 during their active phase (at night). On day 1, rats were habituated for 10 min to an open-field Plexiglas arena (45 cm × 45 cm × 45 cm) in the absence of objects. On day 2, they underwent a 10-min familiarization phase for NOR with identical objects located in opposite corners. On day 3, they underwent the test phase, in which long-term memory at 24 h was examined. During this phase, one familiar (F) object was replaced by a novel one (N), and rats were allowed to explore the arena for 10 min [33]. The objects were of different shapes, colors and textures, and they were thoroughly cleaned with 1% acetic acid between animals to ensure the absence of any olfactory cues. All sessions were videotaped and manually scored by a blind experimenter. The time of exploration of the N and F objects was recorded. The difference in the exploration time of N and F objects was calculated using the discrimination index (DI) as (time N − time F)/(time N + time F).

### 2.4. Magnetic Resonance Imaging (MRI)

Animals were scanned using a 7-Tesla Biospec 70/20 scanner (Bruker, Ettlingen, Germany). A coronal T2-weighted spin-echo sequence was acquired with TE = 33 ms, TR = 3732 ms, averages 2 and slice thickness 0.4 mm. Matrix size was 256 × 256 pixels at a FOV of 3.5 × 3.5 cm^2^ [12,13,14]. T2 images were registered to a common spatial reference using the rigid registration algorithms described in [34]. Then, the hippocampus was manually segmented on each MRI image, according to [35].

### 2.5. Genomic DNA Extraction and 16S Ribosomal RNA Sequencing for Metagenomics

Frozen tissue samples (−80 °C) from the intestinal cecum were used for the study of microbiota. Genomic DNA (gDNA) was extracted from 200 mg of frozen cecum feces using E.Z.N.A.^®^ DNA Stool Kit (VWR, Madrid, Spain) and provided at least 200 μL of genomic DNA. These gDNA samples were then quantified using a BioPhotometer^®^ (Eppendorf, Madrid, Spain) and their concentrations were finally diluted to 6 ng/μL. The diluted samples were used for performing polymerase chain reaction (PCR) amplification, following the protocol of the Ion 16 Metagenomics kit (Thermo Fischer Scientific, Madrid, Spain) [36].

PCR amplification products were used to create a library using the Ion Plus Fragment Library kit for AB Library Builder System (Thermo Fischer Scientific, Madrid, Spain), with sample indexing using the Ion Xpress Barcode Adapters 1–96 kit (Thermo Fischer Scientific, Madrid, Spain).

Template preparation was performed using the ION OneTouch 2 System and the ION PGM Hi-Q OT2 kit (Thermo Fischer Scientific, Madrid, Spain). Metagenomics sequencing was performed using the ION PGM Hi-Q Sequencing kit (Thermo Fischer Scientific, Madrid, Spain) on the ION PGM System. The chips used were the ION 314 v2, 316 v2 or 318 v2 Chips (Thermo Fischer Scientific, Madrid, Spain) with various barcoded samples per chip.

### 2.6. Phylogenetic Analysis

The consensus spreadsheet for each metagenomics sequencing was downloaded from ION Reporter software (version 5.6, Life Technologies Holdings Pte Ltd., Singapore). This spreadsheet included the percentages for each taxonomic level and was used for comparing frequencies between individuals and groups. Analyses were carried out using QIIME-2 software (version 2017.6.0).

### 2.7. Statistical Analysis of Data

Normality and homoscedasticity of each variable were tested using Shapiro-Wilk’s and Levene’s test, respectively. Data were analyzed by means of 2-way ANOVAs (considering MIS and NAC7 or MIS and NAC21 as factors) followed by Tukey posthoc test or Kruskal–Wallis analysis followed by Dunn’s multiple comparison test when normality and homoscedasticity were not met. A *p* value < 0.05 was considered statistically significant. When a trend was observed between control animals (Saline-VH) and MIS-VH animals, a Mann–Whitney or *t*-test was also performed. Analysis of the microbiome community was carried out by non-supervised multivariate analysis (PCA) using R software (v3.2.4). For LDA analysis, tab-delimited files were generated in GraphPad and computed at the family level. The Kendall rank correlation coefficient was used to measure the ordinal association between gut microbiota and hippocampal volume. Graphic representations of all the data were generated with GraphPad Prism software (version 8, GraphPad Software, San Diego, CA, USA). Data were expressed as mean ± standard error of the mean (SEM). Whenever results are statistically significant, it is indicated in the figures.

## 3. Results

### 3.1. Memory

Two-way ANOVA analysis revealed that NAC led to significant differences in the discrimination index for long-term memory (*p* < 0.01) (Figure 1A). Thus, there was a significant reduction in the discrimination index in MIS-VH animals (pathological animals without NAC treatment) relative to Saline-VH (healthy control animals without NAC treatment) (*p* < 0.05), which was prevented by NAC treatment from the immune stimulation to delivery (*p* < 0.05). In addition, an interaction between Phenotype and NAC21 was found (*p* < 0.05).

### 3.2. Hippocampal Volumetric Changes

ANOVA analysis of hippocampal volume measurements showed a significant effect of MIS in this brain area, with reduced volume in MIS-VH versus the Saline-VH group (*p* < 0.01) (Figure 1B). In addition, a significant effect of NAC treatment was found, with increased hippocampal volume in MIS NAC-treated animals (7 and 21 d) (*p* < 0.01).

### 3.3. Microbiota Changes in MIS Offspring

ANOVA analysis of bacterial richness as measured by the Shannon index (Figure 1C) showed a significant effect of MIS (*p* < 0.05) and NAC (*p* < 0.01), with reduced bacterial richness in MIS-VH compared to Saline-VH animals (*p* < 0.05). NAC treatment during pregnancy increased bacterial richness in MIS animals, regardless of whether NAC was administered throughout gestation or from the immune stimulation to delivery.

ANOVA analysis of bacterial diversity as measured by the Chao index (Figure 1C) showed a significant effect of NAC (*p* < 0.05). No significant effect of MIS or any interaction was found (*p* > 0.05).

At the phylum level (Figure 2, Appendix A), *Bacteroidetes* and *Firmicutes* populations were the most abundant phyla in all groups. *Proteobacteria* was the third most common phylum, followed by *Actinobacteria*. Statistical analysis found differences between groups in the percentage of *Firmicutes*, *Firmicutes/Bacteroidetes* ratio, *Proteobacteria* and *Deferribacteres* (Appendix A). Posthoc analysis showed a *Proteobacteria* population increase in the Saline-NAC 21d group vs. Saline-NAC 7d (*p* < 0.05). NAC treatment during 21 d increased *Deferribacteres* in both Saline and MIS animals. In addition, the analysis of the *Firmicutes*/*Bacteroidetes* ratio, which has been suggested as an important index for healthy gut microbiota [34], showed a reduced ratio in MIS-NAC7 compared to MIS-VH (*p* < 0.09). No other statistical differences were observed in any other phyla (Appendix A).

At the family level (Figure 3), in the control group (Saline-VH), the most abundant families were *Lachnospiraceae* (30.7%), *Porphyromonadaceae* (13.7%), *Clostridiaceae* (12.2%), *Prevotellaceae* (11.9%) and *Ruminococcaceae* (11.3%). In the MIS-VH group, the most abundant ones were *Lachnospiraceae* (23.7%), *Clostridiaceae* (12.9 %), *Ruminococcaceae* (12.4%), *Prevotellaceae* (11.0%), *Lactobacillaceae* (10.9%), *Porphyromonadaceae* (9.1%) and *Bacteroidaceae* (9.0%). These proportions were almost unchanged by NAC treatment (7 d and 21 d) except for *Prevotellaceae,* which increased with NAC treatment, *Lactobacillaceae* and *Bacteroidaceae*, which decreased with NAC in a dose-dependent manner.

These compositions at a family level were relatively homogeneous between animals in each group, as can be seen in Appendix A.

Likewise, at the family level (Appendix A), statistically significant differences between groups were found in several taxa, including the *Firmicutes* taxon *Clostridiales XVI* family, which is strongly reduced in the MIS-VH vs. Saline-VH group (*p* < 0.001), although NAC treatment during whole pregnancy restored this reduction (Figure 4). For *Veillonellaceae*, there was a statistically significant reduction in MIS-VH vs. Saline-VH animals (*p* < 0.05), with a certain tendency for its restoration under NAC treatment in MIS animals. *Erysipelotrichaceae* populations were strongly reduced in MIS-VH animals, but NAC for 21 d increased these levels significantly (*p* < 0.05). The *Lactobacillaceae* family showed a significant increase in MIS-VH vs. Saline-VH groups (**p* < 0.05, ** *p* < 0.01).

As for the families of the phylum *Bacteroidetes*, the MIS challenge significantly affected two families. *Bacteroidaceae* populations were strongly increased in MIS-VH vs. Saline-VH animals (*p* < 0.01), an effect that was restored to normal conditions with NAC treatment in a dose-dependent manner. The *Rikenellaceae* family was reduced in MIS-VH vs. Saline-VH group, and NAC treatment restored these populations. With respect to *Proteobacteria* families, the *Pasteurellaceae* taxon increased in MIS-NAC21 vs. MIS-VH animals. Finally, regarding the *Actinobacteria* phylum, the *Corynebacteriaceae* taxon was not present in the MIS groups. Additionally, in the case of *Bifidobacteriaceae*, no populations were detected in the Saline groups but increased in MIS-VH animals, while these populations were strongly reduced in the MIS-NAC7 and MIS-NAC21 groups.

At the genus and species level (Figure 5 and Appendix A), statistically, significant differences were mainly found in *Firmicutes* phylum taxa. *Lactobacillus* was increased in MIS-VH vs. Saline-VH animals, and the same phenomenon occurred in *L. intestinalis* and *L. kitasatonis* species. Importantly, NAC7 and NAC21 treatments were able to reduce these increases in MIS animals, although the statistical significance of this reduction was only observed in the *L. kitasatonis* MIS-NAC21 group. *Acetatifactor* populations were strongly reduced in MIS-VH vs. Saline-VH animals, but both NAC treatments prevented this reduction. The genus *Lachnoanaerobaculum* was also reduced in MIS animals, and, as above, NAC7 treatment returned this decrease to normal levels. The behavior of *Roseburia* and *Turicibacter* was similar, with an increase in their populations in the MIS animals with NAC treatments, above the ones in Saline animals.

For the phylum *Bacteroidetes*, in general, MIS-VH animals showed a strong increase in populations of the genus *Bacteroides* (such as in *B. uniformis*), which showed a tendency to decrease in a dose-dependent manner with NAC treatments.

In the phylum *Proteobacteria*, no *Desulfovibrio* populations were detected in MIS-VH animals, but NAC increased them in a dose-dependent manner in both Saline and MIS animals.

Finally, in the phylum *Actinobacteria*, the species *Corynebacterium stations* disappeared in MIS-VH animals and did not recover with NAC treatments.

### 3.4. Principal Component Analysis of the Bray-Curtis Distance of the Microbiota

To determine whether overall gut microbiome composition differed between groups, we performed a principal component analysis (PCA) of the Bray-Curtis distance (Figure 6A). The PCA1 and PCA2 components explained 29% of the variance between the different animal groups. We found that MIS-VH animals presented the highest data dispersion compared to the other groups, indicating differences in the gut microbiota composition associated with the maternal immune challenge. Indeed, those animal groups receiving NAC treatment, regardless of its duration (7 or 21 days), showed a distribution of microbiota taxa more similar to that of the healthy animals’ group (Saline_VH) than to the pathological group (MIS_VH), thus suggesting that NAC treatment was able to return the variability of gut microbiota composition to similar parameters in MIS animals as in Saline animals.

### 3.5. Linear Discriminant Analysis of the Microbiota

Bacterial families with significant differences in their relative abundances between Saline and MIS animals are indicated in the linear discriminant analysis (LDA) (Figure 6B).

The main bacterial families that better discriminate between healthy and pathological controls (Saline_VH and MIS_VH) include *Pseudanabaenaceae, Enterococcaceae* and *Leuconostocaceae* (among others) for MIS_VH group, and *Oceanospirillaceae, Carnobacteraceae* and *Kiloniellaceae* (among others) for Saline_VH group. The main bacterial family that better discriminated between those groups that were treated with NAC during 21 days is *Paenibacillaceae*, which belongs to the *Firmicutes* phylum.

### 3.6. Kendall Correlation between Gut Microbiota Taxa and Hippocampal Volume

The taxa associated with MIS were negatively correlated with hippocampal volume (*Thermoanaerobacteraceae*) at a trend level (*p* = 0.06). 7-day and 21-day treatment with NAC during pregnancy-induced changes in the different taxa in control and MIS animals. Thus, the taxa associated with controls from dams treated with NAC for seven days were negatively correlated with hippocampal volume (*Lachnospiraceae*, *p* < 0.05), whereas in MIS animals, a positive correlation was found in *Lachnospiraceae* (*p* < 0.05), and a negative correlation was found in *Sutterellaceae* (*p* < 0.05). Moreover, NAC treatment during the whole pregnancy was negatively correlated with the hippocampal volume in control animals (*Pasteuralleaceae*, *p* < 0.05; *Veillonellaceae*, *p* < 0.05) and MIS animals (*Sphingobacteriaceae*, *p* < 0.01).

## 4. Discussion

In this study, we aimed to describe the effect of NAC administration during pregnancy on the gut-brain axis in the offspring of the MIS animal model of schizophrenia. First, we evaluated changes in hippocampal size and long-term memory after NAC treatment. Second, we evaluated changes in bacterial composition, taxonomy, and functional diversity of the gut microbiota. Finally, we analyzed the potential correlation between microbiome and hippocampal data.

The present study confirmed: (1) the existence of gut microbiota alterations in the offspring from Poly I:C-treated dams similar to those found in patients with schizophrenia, such as lower bacterial richness and an increase in *Lactobacillaceae*, *Bifidobacteriaceae* and *Bacteroidaceae* families; (2) the efficacy of NAC treatment during pregnancy to prevent hippocampal volume reduction, long-term memory deficits, and partially recover the alterations in the gut microbiota. Therefore, changes in brain volume, behavior, and microbiome in MIS offspring may be mediated by reduced oxidative stress levels, which led to an attenuated inflammatory response due to NAC treatment.

In recent years, much attention has been paid to the bidirectional homeostasis between the gut and the brain [37]. Communication between them involves neuroimmunoendocrine mediators through the central, autonomic, and enteric nervous systems and the hypothalamic-pituitary-adrenal (HPA) axis [38]. Gut dysbiosis, i.e., the imbalance in the proportions of the different elements that compose the bacterial microbiome, has been associated with the onset of different psychiatric and neurodevelopmental disorders, such as schizophrenia [38]. Furthermore, these patients may also display an increased intestinal permeability [38]. To date, nearly a dozen studies have reported differences in the microbiome between healthy individuals and patients at different stages of the disease [39,40]. Nevertheless, their results are not consistent, especially when analyzing bacterial components in a higher taxonomic order.

As expected, we found lower gut microbiota diversity in MIS animals compared to healthy animals; however, no changes in gut microbiota richness were found. In schizophrenia, results are not consistent so far. Recent studies reported disturbances in gut bacterial taxon composition, with lower gut microbiome diversity index, leading to lower numbers and species [41], but its clinical significance is not well described [42]. Nevertheless, a recent meta-analysis showed that microbial diversity was well preserved [43]. In terms of the gut microbiota richness index, this has been found to be generally low in patients with schizophrenia [42,43]. Microbial richness is involved in many cellular processes, such as the inflammatory response or those related to the maintenance of cell junction integrity. Thus, alterations in the richness index may suggest a weakness in epithelial barrier functions [44]. In our study, microbial richness was reduced in MIS-offspring, suggesting that this rat model may present altered epithelial barrier function, although further studies are needed to corroborate this fact. In addition, we found an effect of NAC treatment on both diversity and richness indices, increasing both indices in MIS-treated animals and suggesting a beneficial effect of NAC treatment on maintaining a healthy, resilient gut.

At the phylum level, *Firmicutes* and *Proteobacteria* taxa have shown consistent alterations in patients with schizophrenia [45]. Furthermore, the *Firmicutes:Bacteroidetes* (*F:B*) ratio, accepted as an important index in the normal intestinal homeostasis for the two dominant phyla in our gut microbiome, is also altered in schizophrenia, with enrichment of *Firmicutes* members and a decrease of *Bacteroidetes* [45,46]. In our study, we found no differences in the *F:B* ratio between healthy and MIS animals, as has been noted by other authors [15]. Nonetheless, although an increased *F:B* ratio has been linked to schizophrenia, some studies suggest that its alteration could be due to the metabolic disorder induced by antipsychotic medication since disorders, such as obesity or diabetes, in which inflammation and oxidative stress play an important role, also show an altered *F:B* ratio [47,48,49].

At the family taxa, the most consistent alteration in patients with schizophrenia is enrichment within the *Lactobacillaceae* family [37]. This imbalance has been associated with the evolution of the disorder and the severity of the symptoms, especially in the first episode of psychosis [50]. Interestingly, we successfully corroborated this finding in our MIS animals, and remarkably, NAC treatment during pregnancy prevented this enrichment. Specifically, we observed changes in the following species: *Lactobacillus animalis*, *L. johnsonii*, *L. kitasatonis*, *L. murinus*, *L. reuteri* and *L. vaginalis* (see Appendix A). This result is somehow surprising, especially considering that *Lactobacillus* has traditionally been suggested as a probiotic-based treatment. This apparent contradiction could be explained by the fact that *Lactobacillus*, as a probiotic treatment, does not colonize the digestive tract, as does the commensal microbiota.

The *Bacteroidaceae* family has also been extensively characterized in this disorder. Although some discrepancies are observed between different studies, most authors reported an increase in this bacterial family in patients with schizophrenia at different stages of the disease [37,50]. In our study, we corroborated this increase in *Bacteroidaceae* in MIS-offspring, and again, NAC treatment successfully prevented this effect, especially when administered for 21 days. Furthermore, we surprisingly observed differential effects of NAC on *Clostridiales*, *Veillonellaceae* and *Lachnospiraceae* families in control and MIS animals, with a reduction of bacteria from these families in healthy animals and an enrichment in MIS animals. These results could suggest a differential modulatory effect of NAC as a function of the different inflammatory and oxidative statuses in the animal. Interestingly, NAC7 and NAC21 treatments resulted in a reduction of proinflammatory bacterial taxons in MIS animals, such as in the case of reduced *Bilophila wadsworthia*, or in an increase in taxons known to produce anti-inflammatory metabolites (e.g., short-chain fatty acids), such as the genera *Acetatifactor*, *Roseburia*, *Lachnoaerobaculum* or *Turicibacter*; and families, such as *Prevotellaceae.* This finding is of particular relevance considering the key role of inflammation and oxidative stress in the pathogenesis of schizophrenia [51] and in our rat model [14]. Thus, changes in the microbiome are probably mediated by reduced oxidative stress levels due to NAC treatment, which led to an attenuated inflammatory response due to NAC treatment. Nevertheless, further studies evaluating the ability of NAC to modulate direct inflammatory markers are needed to fully demonstrate its anti-inflammatory role.

As stated above, regulation of the HPA axis is influenced by the state of the gut microbiome. Thus, alterations in different components of the HPA axis and the bacterial microbiota may reciprocally modulate each other [52]. Furthermore, alterations of this neural axis have been repeatedly linked to the schizophrenia pathophysiology, although its exact role in the course of this disorder remains to be clarified [53]. In this sense, one important brain region involved in the inhibitory regulation of this axis is the hippocampus, which is remarkably altered through changes in the microbiota, as demonstrated by Tang et al. [54]. In addition, the hippocampus is functionally and anatomically altered in patients with schizophrenia, showing a reduction in the volume of this structure in most patients [55,56]. Of relevance, a similar reduction in hippocampal volume has been found in animals of the MIS model induced by the viral mimetic Poly I:C [10,11,12,13,14]. Here, we replicated this hippocampal reduction and, more importantly, we demonstrated, for the first time, that NAC treatment during pregnancy was able to fully prevent this alteration. Given that inflammation in the hippocampus is key to vulnerability and recovery from psychiatric disorders. [54], these results are of great interest. On the other hand, the hippocampus is the center of learning and memory [57]. In this regard, schizophrenia is associated with cognitive deficits, including problems with short- and long-term working memory [58]. Prenatal Poly I:C challenge also affects behavior at the memory level, as previously demonstrated by our group and others [33,59]. In our setting, MIS animals showed memory impairments in the novel object recognition task, with a significant reduction in the object discrimination index for long-term memory, which was prevented by NAC treatment from the immune stimulation to delivery, although a similar trend was found for NAC treatment throughout pregnancy. Our results suggest that microbiota-based interventions with anti-IOS compounds could potentially be applied to prevent hippocampal-based memory impairments in neuropsychiatric disorders.

Lastly, gut microbiota composition has been associated with the severity of psychotic symptoms [60], but no relationship between brain structural changes by means of MRI and gut microbiota after NAC treatment has been described. Here, we found some correlations between hippocampal volume and gut microbiome after NAC treatment during pregnancy. Particularly, we found a correlation in the *Lachnospiraceae* taxa in the offspring from dams treated with NAC for seven days. Importantly, this taxon has been associated with the onset of negative symptoms in patients with schizophrenia [3,61] and is known to increase after antipsychotic treatment [4,62]. Thus, the positive correlation of *Lachnospiraceae* taxa could be related to the beneficial effect of NAC treatment in reducing the inflammatory basis of the MIS model. In addition, we also found a negative correlation between hippocampal volume and *Sphingobacteriaceae* taxa in MIS offspring from dams treated with NAC for 21 days. Alterations in this family have been linked to the pathogenesis of schizophrenia and bipolar disorder [63], as well as other inflammatory-based diseases, such as inflammatory bowel disease (IBD) [64,65], in which anxiety and depression are also present, and these are two distinctive features also described in the MIS model [66,67]. Of note, a large majority of cognitive impairments in the MIS model are possible through changes in the hippocampal function [68]. In this regard, we showed the existence of a reduced hippocampal volume in MIS animals, which may be responsible for the hippocampal-based memory impairment in the object discrimination index. Thus, an improvement in the hippocampal size would correlate with a reduction in the *Sphingobacteriaceae* taxa.

Finally, this study had some limitations. First, the small number of animals probably restricted statistical power, especially in the correlation tests. We believe that, with a larger number of animals, more statistically significant differences could have been found. Second, the study suffers from the inherent limitations of any psychiatry animal model. In this sense, the MIS model is considered a valid model for neurodevelopmental disorders, but we are aware that no animal model can fully mimic the specific characteristics of this type of pathology in humans. Third, we did not correct the statistical analyses for multiple comparisons to prevent type I errors, but the exploratory nature of this study warranted the suitability of this methodological choice. This is a common practice in studies of an exploratory nature, as in our case, and also noted by other authors [69]. Future dedicated studies are necessary to validate our findings. Finally, this study has only been conducted in males. Future studies, including both males and females, are warranted.

## 5. Conclusions

The current state of research on the microbiome in human disorders is at an extremely early stage. There are few certainties and large discrepancies in the existing data, probably due to the still low number of studies and the heterogeneity of the methodologies used. Beyond the few certainties we currently have, what does seem evident is that there is a relationship between alterations in the microbiome and the course of schizophrenia. Our study confirmed the existence of gut microbiota alterations in the offspring from Poly I:C-treated dams, similar to those found in patients with schizophrenia. Interestingly, these gut microbiota alterations were partially prevented by NAC treatment during pregnancy. In addition, we demonstrated the efficacy of NAC treatment in preventing hippocampal volume reduction and long-term memory deficits. These results suggest that early approaches with anti-IOS compounds, especially in neurodevelopmental disorders with an inflammatory/oxidative basis, may be useful in modulating bacterial microbiota, hippocampal size and hippocampal-based memory impairments.

## Figures and Tables

**Figure 1 antioxidants-12-00970-f001:**
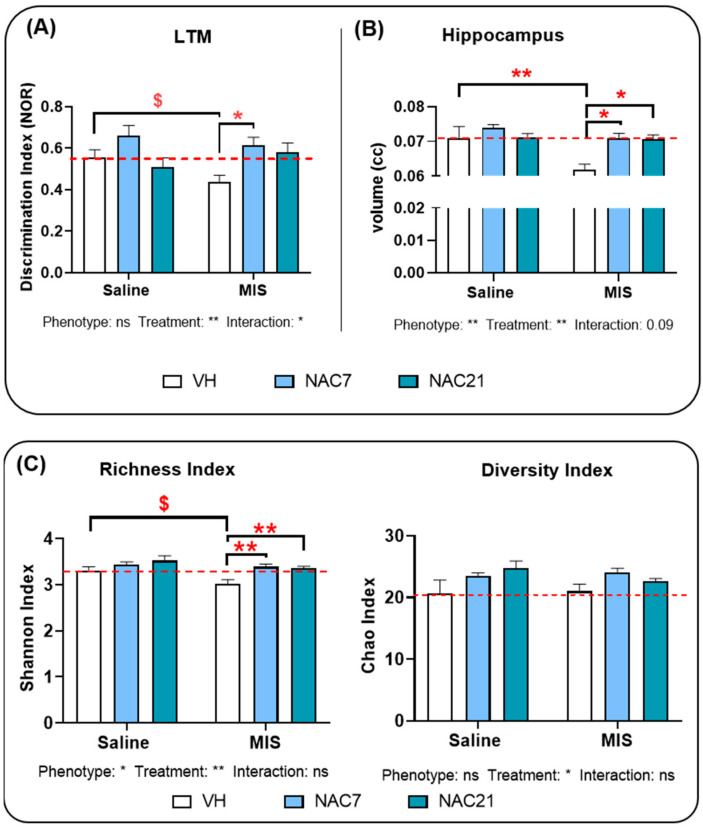
(**A**) **Discrimination index.** The effect of NAC treatment on the long-term memory (LTM) discrimination index between objects in the novel object recognition test (NOR) (Saline VH, N = 9; Saline NAC7, N = 6; Saline NAC21, N = 8; MIS VH, N = 7; MIS NAC7, N = 8; MIS NAC21, N = 6). (**B**). **Hippocampal volume.** The effect of NAC on hippocampal volume in the Saline (control) and MIS (pathological) animal groups, measured by MRI. (Saline VH, N = 7; Saline NAC7, N = 12; Saline NAC21, N = 12; MIS VH, N = 6; MIS NAC7, N = 12; MIS NAC21, N = 12). (**C**). **Bacterial diversity and richness**. Representation of the changes in bacterial diversity and richness, as measured by Shannon and Chao Indexes in Saline and MIS animals after a preventive treatment with NAC during pregnancy. (Saline VH, N = 8; Saline NAC7, N = 11; Saline NAC21, N = 11; MIS VH, N = 11; MIS NAC7, N = 10; MIS NAC21, N = 9). [Data is shown as mean ± SEM. 2-way ANOVA followed by a posthoc test, Tukey’s multiple comparison (* *p* < 0.05, ** *p* < 0.01) and unpaired *t*-test (^$^ *p* < 0.05) are shown. ns: non-significant]. The red dotted line indicates the mean value of the control group (Saline-VH).

**Figure 2 antioxidants-12-00970-f002:**
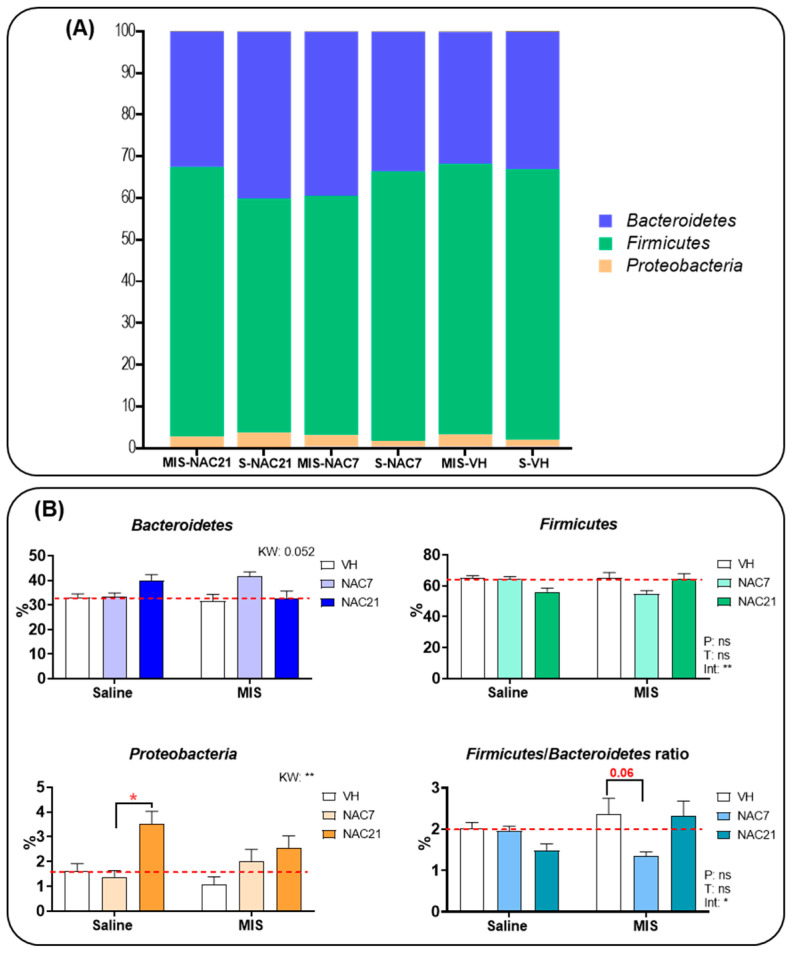
(**A**) **The nested bar at the phylum level.** The nested bar graph shows the mean percentage composition of the most representative phyla for each group. Bacteroidetes (purple), Firmicutes (green) and Proteobacteria (orange) were the most common phyla. (**B**) **Gut microbiota composition at the phylum level.** Graphs show the mean percentage composition of the 3 most representative phyla and the Firmicutes/Bacteroidetes ratio in the Saline and MIS animals treated with NAC during gestation. (Saline VH, N = 8; Saline NAC7, N = 11; Saline NAC21, N = 11; MIS VH, N = 11; MIS NAC7, N = 11; MIS NAC21, N = 11). Data are shown as mean ± SEM. 2-way ANOVAs followed by Tukey posthoc test or Kruskal–Wallis (KW) analysis followed by Dunn’s multiple comparison are shown (* *p* < 0.05, ** *p* < 0.01) [ns: non-significant, P: Phenotype, T: Treatment, int: Interaction]. The red dotted line indicates the mean value of the control group (Saline-VH).

**Figure 3 antioxidants-12-00970-f003:**
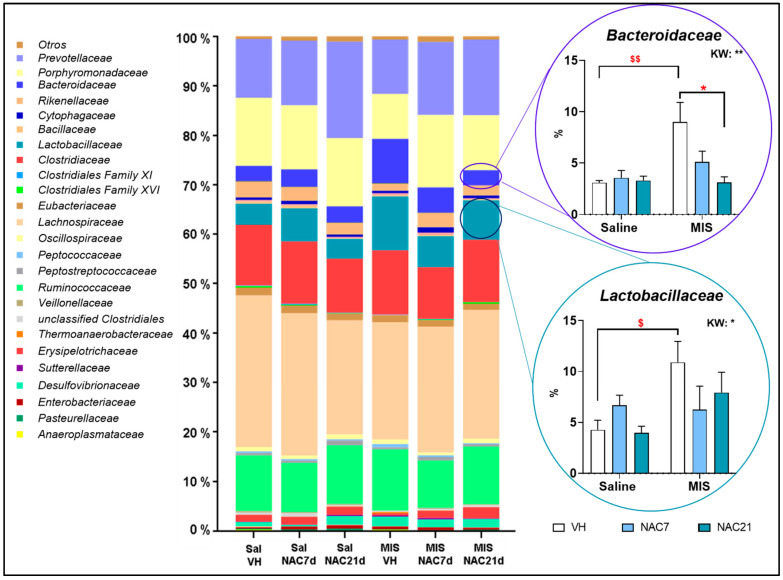
**Nested bar plot showing gut microbiota composition at the family level**. Differences in average intestinal microbiota composition at the family level. The nested bar graph shows the mean percentage composition of the most representative families for each group. A higher proportion of Bacteroidaceae and Lactobacillaceae families was found in MIS-VH animals compared to the control group (Sal-VH), which was prevented by NAC treatment. Kruskal–Wallis (KW) analysis followed by Dunn’s multiple comparison test (* *p* < 0.05, ** *p* < 0.01) and Mann–Whitney test (^$^ *p* < 0.05, ^$$^ *p* < 0.01) are shown.

**Figure 4 antioxidants-12-00970-f004:**
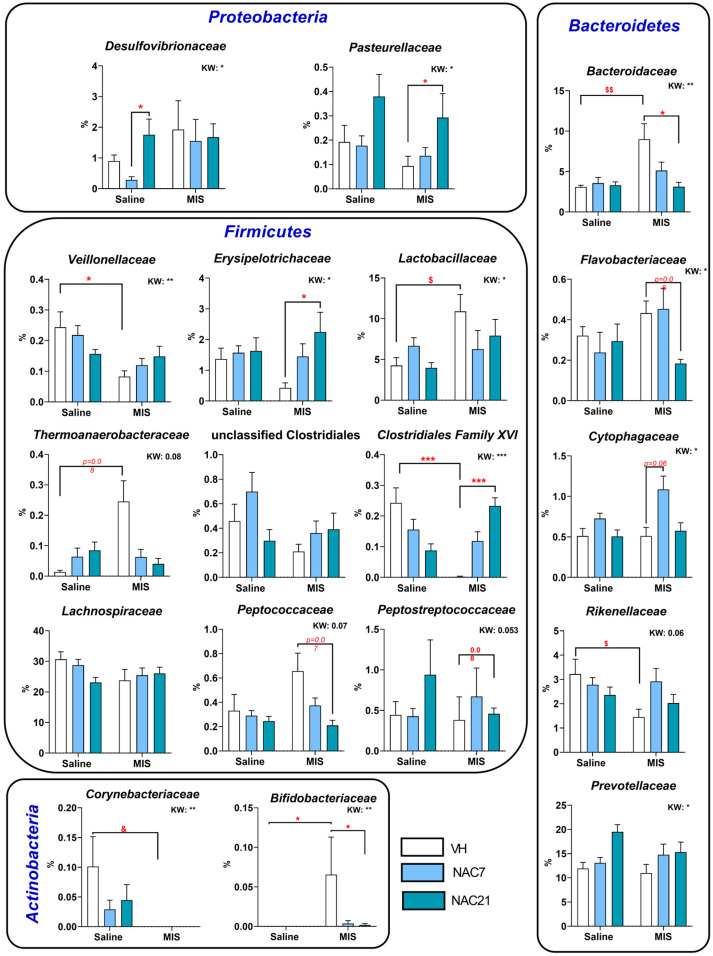
Gut microbiota composition at the family level. Selection of the most representative microbiota families altered in the MIS model and modified by NAC treatment. Each column shows the average of % changes in the composition of microbiota at the family level. (Saline-VH: 8, Saline-NAC7: 11, Saline-NAC21: 11, MIS-VH: 11, MIS-NAC7: 11, MIS-NAC21: 11). Data is shown as mean ± SEM. Kruskal–Wallis (KW) analysis followed by Dunn’s multiple comparison test (* *p* < 0.05, ** *p* < 0.01, *** *p* < 0.001) and Mann–Whitney test (^$^ *p* < 0.05, ^$$^
*p* < 0.01) are shown.

**Figure 5 antioxidants-12-00970-f005:**
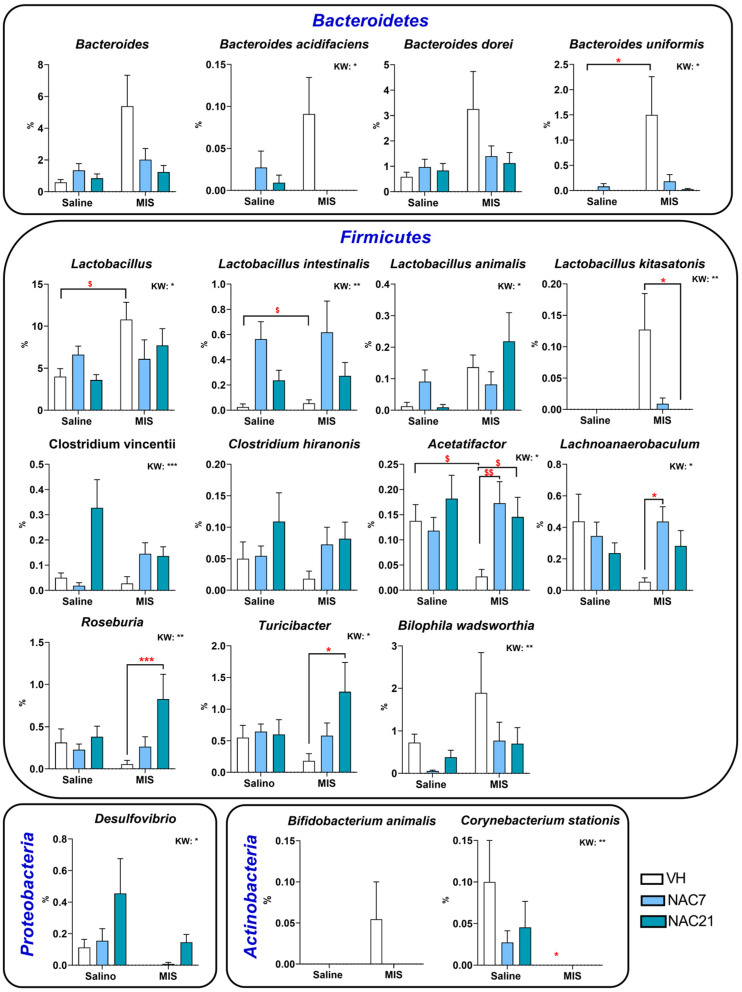
Selection of the most representative genus and species altered in the MIS model and modified by NAC treatment. Each column shows the average of % changes in the composition of microbiota at the genera and species level. Data are shown as mean ± SEM. Kruskal–Wallis (KW) analysis followed by Dunn’s multiple comparison test (* *p* < 0.05, ** *p* < 0.01, *** *p* < 0.001) and Mann–Whitney test (^$^ *p* < 0.05, ^$$^ *p* < 0.01) are shown.

**Figure 6 antioxidants-12-00970-f006:**
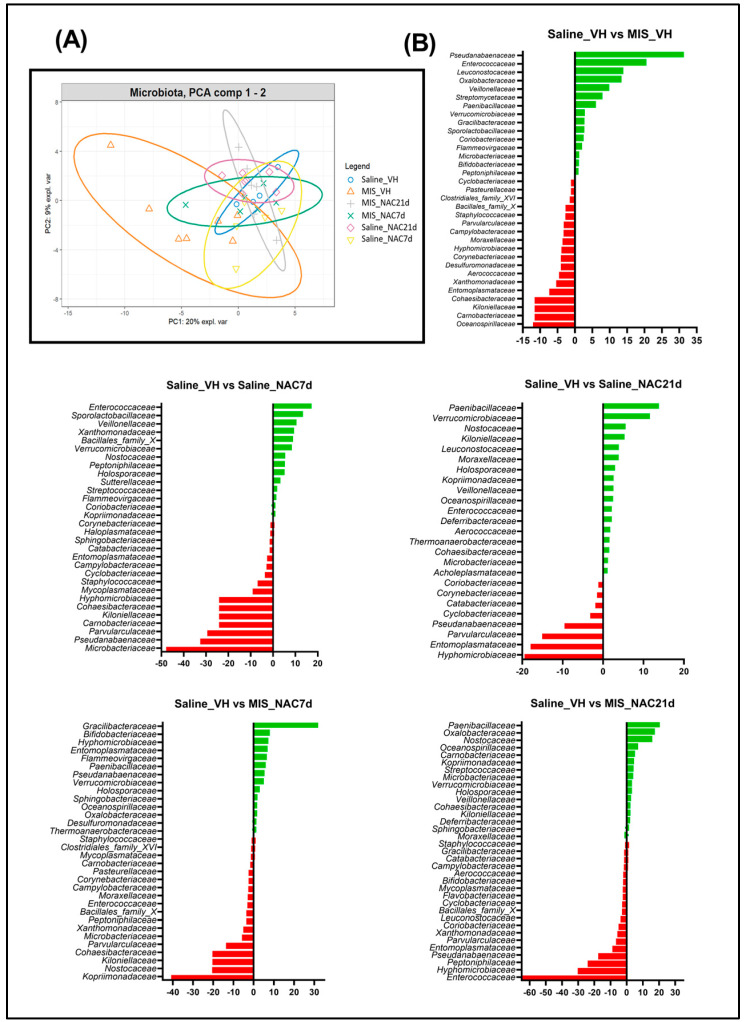
**PCA and LDA analysis**. (**A**). PCA cluster analysis of gut microbiota, which explains the 29% of the observed gut microbiota composition variance among groups. (**B**). LDA analysis among pairs of the different experimental groups, showing the families that better discriminate between the groups (Red: Saline-VH animals, Green: rest of groups).

## Data Availability

Not applicable.

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
