# Peer review of "Maternal Supplementation with N-Acetylcysteine Modulates the Microbiota-Gut-Brain Axis in Offspring of the Poly I:C Rat Model of Schizophrenia"

_antioxidants, 2023, doi:10.3390/antiox12040970_

Round 1

Reviewer 1 Report

This is a perfect work, in my opinion, and its immediate publication is indicated.

Author Response

We thank Reviewer 1 for the revision of the manuscript.

Reviewer 2 Report

I have read this paper with great interest, and value the efforts and paper as reported. I do have two issues that i would like to suggest the authors to further reflect on

is it correct that no power calculation has been made a priori ? 

the authors suggest causality, while perhaps association is a better approach, as it seems that 'modulation' of the intestinal flora is assumed to be the causal mechanism, while modulation of inflammation by NAC can also result in both a different phenotype and changes in microbiome mediated by a blunded inflammatory response as mechanism. 

Author Response

I have read this paper with great interest, and value the efforts and paper as reported. I do have two issues that I would like to suggest the authors to further reflect on

is it correct that no power calculation has been made a priori? 

The number of animals per cohort was calculated for the six experimental groups, considering the three Rs principle in animal experimentation and assuming independent groups, the detection of large effect size (1 SD), and a power of 80%. This information has been added in the methods section. In addition, we have included the effect size in the supplementary tables.

the authors suggest causality, while perhaps association is a better approach, as it seems that 'modulation' of the intestinal flora is assumed to be the causal mechanism, while modulation of inflammation by NAC can also result in both a different phenotype and changes in microbiome mediated by a blunted inflammatory response as mechanism. 

The reviewer is right. Our intention was to modulate the inflammation that has been reported in this animal model.

Reviewer 3 Report

In this paper, Romero-Miguel and colleagues present tests looking at the effect of N-acetylcysteine (NAC) on the microbiota of rats whose mother had her immune system activated by poly-I:C during pregnancy, which is established to cause schizophrenia-like symptoms.  Based on the introduction, the idea here seems to be to establish in NAC may improve schizophrenia symptoms in part by altering the microbiota, as part of its anti-inflammatory effect. The idea is interesting, but has several major drawbacks.

Firstly, the number of animals used is very low, as acknowledged by that authors. How were these numbers chosen?  Were power calculations performed to determine an optimal number?

Secondly, and related to this, more clarification is needed in how multiple testing was accounted for in the statistics throughout the manuscript.  In the methods section, the authors list that they did a Bonferroni post-hoc test, but it is not clear how many comparisons were included in each experiment. For example, in 1b, were comparisons only done of MIS-VH with the other MISs and with Saline-VH, or was every combination analyzed?  This is unclear, and no results are labelled as non-significant, making it hard to tell where comparisons were or were not made. In figure 3, for example, where 18 microbiota families are examined, are all of these taken into account in the calculations, or is each family investigated individually? Given the sheer number of things examined, in a relatively small number of animals, it is hard to imagine any significant effects being detected.

Thirdly, the authors state in the title that NAC promotes an “anti-inflammatory gut microbiota”. While there are changes in the microbiota, I cannot find anywhere where the authors establish that this change is anti-inflammatory. Certainly no experiments seem to specifically compare changes in “anti-inflammatory gut microbiota” to general gut microbiota.

This leads into the final point, that it is just not clear what these results mean.  Yes, NAC affects the microbiota of these rats, but there is nothing to establish whether this is in any way related to its potential as an adjunctive therapy for schizophrenia.  No behavioral tests are performed, so it is unclear if the microbiota tests correlate with any symptoms. NAC affects various microbiota, but it is not clear that there is an established effect of normalizing the microbiota to that in animals that did not undergo maternal immune activation. There is no alternative schizophrenia model tested that might show similar changes in microbiota.

In summary, the work that the authors have done is OK, but it is very hard to draw any concrete conclusions beyond that there are some microbiota changes in this specific model, and using this specific treatment.  It is certainly not possible to generalize these findings to how NAC might be useful as a schizophrenia treatment, and based on the introduction this seems to be the authors intentions.

Other points:

The authors refer several times to NAC having a dose-dependent effect in their work, however this is misleading, as only one dosage was used.  I assume they refer to how many days animals received NAC for, however any differences here could refer to the timing of the NAC rather than the total does (is treatment in the first 14 days crucial?)

The first citation given is connected to a statement saying that the microbiota-gut-brain axis is important in schizophrenia, with an implication that it refers to humans.  The paper cited, however instead talks about a depression model in rats (and is, incidentally, written by the authors)

Line 67: The sentence that “NAC is sometimes administered as adjunctive treatment” implies that it is established enough to be used as least semi-regularly in some clinics, however the citations are to clinical trials, and not regarding clinical practice.

The authors need to explain abbreviations like NAC21d in section 3.1, where they use them to describe data for the first time.

In figure legends, authors need to clearly define the difference between # and * for p-values, and how they relate to what was tested.

Author Response

In this paper, Romero-Miguel and colleagues present tests looking at the effect of N-acetylcysteine (NAC) on the microbiota of rats whose mother had her immune system activated by poly-I:C during pregnancy, which is established to cause schizophrenia-like symptoms.  Based on the introduction, the idea here seems to be to establish if NAC may improve schizophrenia symptoms in part by altering the microbiota, as part of its anti-inflammatory effect. The idea is interesting, but has several major drawbacks.

Firstly, the number of animals used is very low, as acknowledged by that authors. How were these numbers chosen?  Were power calculations performed to determine an optimal number?

The number of animals per cohort was calculated for the 6 experimental groups, considering the 3Rs principle in animal experimentation and assuming independent groups, the detection of large effect size (1 SD) and a power of 80%. This information has been added in the methods section. In addition, we have included the effect size in the supplementary tables.

Secondly, and related to this, more clarification is needed in how multiple testing was accounted for in the statistics throughout the manuscript.  In the methods section, the authors list that they did a Bonferroni post-hoc test, but it is not clear how many comparisons were included in each experiment. For example, in 1b, were comparisons only done of MIS-VH with the other MISs and with Saline-VH, or was every combination analyzed?  This is unclear, and no results are labelled as non-significant, making it hard to tell where comparisons were or were not made. In figure 3, for example, where 18 microbiota families are examined, are all of these taken into account in the calculations, or is each family investigated individually? Given the sheer number of things examined, in a relatively small number of animals, it is hard to imagine any significant effects being detected.

We agree with the reviewer. We used different symbols for the same comparisons. We apologize for this confusion. Now, all statistical comparisons are well described and symbolled. Our study factors were: phenotype (Saline, MIS) and treatment (VH, NAC7 and NAC21). Data were analyzed by means of 2-way ANOVAs followed by Tukey post-hoc test or Kruskal–Wallis analysis followed by Dunn’s multiple comparison when normality and homoscedasticity were not met. When a trend between control animals (Saline-VH) and MIS-VH animals was observed, Mann–Whitney or t- tests were performed. In addition, we have included the effect size in the supplementary tables for a better understanding of the effect detected.

Thirdly, the authors state in the title that NAC promotes an “anti-inflammatory gut microbiota”. While there are changes in the microbiota, I cannot find anywhere where the authors establish that this change is anti-inflammatory. Certainly no experiments seem to specifically compare changes in “anti-inflammatory gut microbiota” to general gut microbiota.

Reviewer is right. Although we have observed some beneficial effects of both NAC treatments in reducing proinflammatory taxa and increasing of taxa known to produce anti-inflammatory metabolites, other beneficial effects have also been found in other families, genera and species. We have therefore modified the title of the manuscript.

In any case, a new paragraph about the regulation of these taxa has been included in the Discussion section.

This leads into the final point, that it is just not clear what these results mean.  Yes, NAC affects the microbiota of these rats, but there is nothing to establish whether this is in any way related to its potential as an adjunctive therapy for schizophrenia.  No behavioral tests are performed, so it is unclear if the microbiota tests correlate with any symptoms. NAC affects various microbiota, but it is not clear that there is an established effect of normalizing the microbiota to that in animals that did not undergo maternal immune activation. There is no alternative schizophrenia model tested that might show similar changes in microbiota.

We agree with the reviewer that it is very difficult to interpret the scope of our results without including any behavioral study. Considering that MIS animals showed reduction of the hippocampus volume, we considered appropriate to study the hippocampal-based memory impairment in this model, measured by the novel object recognition test. The information related to this study has been included in the methods, results and discussion sections.

In summary, the work that the authors have done is OK, but it is very hard to draw any concrete conclusions beyond that there are some microbiota changes in this specific model, and using this specific treatment.  It is certainly not possible to generalize these findings to how NAC might be useful as a schizophrenia treatment, and based on the introduction this seems to be the authors intentions.

Thank you very much of all the comments of this reviewer that have undoubtedly helped to improve this article. We hope that all these changes could help the readers to stablish the scope of our study.

Other points:

The authors refer several times to NAC having a dose-dependent effect in their work, however this is misleading, as only one dosage was used.  I assume they refer to how many days animals received NAC for, however any differences here could refer to the timing of the NAC rather than the total does (is treatment in the first 14 days crucial?)

Reviewer is right. Indeed, we refer to the timing of the NAC treatment rather than the total dose. What really matters here is the timing of initiation of NAC treatment, after the immune challenge (which could be considered as treatment of the IOS triggered by the Poly I;C infection) or throughout pregnancy (which could be considered as preventive as it could reduce the IOS basis in the mother, helping her to be prepared in case of any infection). In any case, more studies are needed to understand how NAC treatment could be affecting the IOS markers in these dams.   

The first citation given is connected to a statement saying that the microbiota-gut-brain axis is important in schizophrenia, with an implication that it refers to humans.  The paper cited, however instead talks about a depression model in rats (and is, incidentally, written by the authors)

We apologize for this mistake. Indeed, this article talks about a model of depression and has been written by a Spanish group, but not by our group. In any case, this paper has been removed and replaced by another more appropriate one.

Line 67: The sentence that “NAC is sometimes administered as adjunctive treatment” implies that it is established enough to be used as least semi-regularly in some clinics, however the citations are to clinical trials, and not regarding clinical practice.

We apologize for this mistake. This sentence has been modified as: “In clinical trials, NAC is sometimes administered as adjunctive treatment to conventional treatment with antipsychotics”.

The authors need to explain abbreviations like NAC21d in section 3.1, where they use them to describe data for the first time.

The terms NAC7d and NAC21d have been included in the methods section.

In figure legends, authors need to clearly define the difference between # and * for p-values, and how they relate to what was tested.

The explanation of all figures has been improved for a better understanding of them.

Round 2

Reviewer 3 Report

The authors have addressed the majority of my concerns, but some still need to be addressed:

1) In graphs, please indicate with “ns” for not significant (or an alternative similar notation), which samples were compared but did not show statistically significant differences

2) In figure 4, when doing statistics on these results, does the multiple testing consider all of the tests performed over all the experiments (across all species of bacterodietes, firmicutes, etc), or only the statistical tests done within each species?  The latter scenario would not be appropriate as, when examining 18 different species, some positive results would be expected by change.

3) The new title is much more appropriate, however I would tweak the English to read “Maternal supplementation with N-acetylcysteine modulates the microbiota-gut-brain axis in offspring of the Poly I:C rat model of schizophrenia”

Author Response

Reviewer 3

The authors have addressed the majority of my concerns, but some still need to be addressed:

1) In graphs, please indicate with “ns” for not significant (or an alternative similar notation), which samples were compared but did not show statistically significant differences

We agree with the reviewer that nonsignificant differences should be acknowledged in some way. However, instead of incorporating the notation "ns" for each comparison, which can lead to unclear graphs, we have added this information in the method section. In addition, we have included a more complete clarification of the statistical analysis in the figures for a better understanding of the comparisons.

2) In figure 4, when doing statistics on these results, does the multiple testing consider all of the tests performed over all the experiments (across all species of bacterodietes, firmicutes, etc), or only the statistical tests done within each species?  The latter scenario would not be appropriate as, when examining 18 different species, some positive results would be expected by change.

We partially agree with the reviewer on this point. While we are aware that numerous statistical tests were performed without any further correction in our study, we considered the exploratory nature of this study as a justification warranting the suitability of this methodological choice. In addition, it should be highlighted that this is a common practice in studies of an exploratory nature, as in our case, and also noted by other authors (DOI: 10.1016/j.athoracsur.2015.11.024). However, we recognize that this as a potential limitation, and acknowledge the need for future dedicated studies to validate our findings. In this sense, we have explicitly recognized this limitation at the end of the manuscript in the limitations section.

3) The new title is much more appropriate, however I would tweak the English to read “Maternal supplementation with N-acetylcysteine modulates the microbiota-gut-brain axis in offspring of the Poly I:C rat model of schizophrenia”

Thanks for this comment. The title has been modified according to the reviewer suggestion.
